# Regulation of Melatonin and Neurotransmission in Alzheimer’s Disease

**DOI:** 10.3390/ijms22136841

**Published:** 2021-06-25

**Authors:** Jaydeep Roy, Ka Chun Tsui, Jonah Ng, Man-Lung Fung, Lee Wei Lim

**Affiliations:** School of Biomedical, Sciences, Li Ka Shing Faculty of Medicine, The University of Hong Kong, Hong Kong, China; jaydeep@connect.hku.hk (J.R.); tsuikc@connect.hku.hk (K.C.T.); u3555462@connect.hku.hk (J.N.); fungml@hku.hk (M.-L.F.)

**Keywords:** melatonin, neurotransmission, Alzheimer’s disease

## Abstract

Alzheimer’s disease is a neurodegenerative disorder associated with age, and is characterized by pathological markers such as amyloid-beta plaques and neurofibrillary tangles. Symptoms of AD include cognitive impairments, anxiety and depression. It has also been shown that individuals with AD have impaired neurotransmission, which may result from the accumulation of amyloid plaques and neurofibrillary tangles. Preclinical studies showed that melatonin, a monoaminergic neurotransmitter released from the pineal gland, is able to ameliorate AD pathologies and restore cognitive impairments. Theoretically, inhibition of the pathological progression of AD by melatonin treatment should also restore the impaired neurotransmission. This review aims to explore the impact of AD on neurotransmission, and whether and how melatonin can enhance neurotransmission via improving AD pathology.

## 1. Introduction

Alzheimer’s disease (AD) is a common neurodegenerative disorder that is associated with advancing age. AD accounts for 50% to 75% of all dementia cases [1], and it is estimated there are about 50 million people with AD worldwide. With an aging global population, it is estimated the number of people with AD will increase to 152 million by 2050 [2]. AD is characterized by the development of amyloid-beta (Aβ) plaques, hyperphosphorylated tau and neurofibrillary tangles (NFT) [3]. The development of these pathologies in the brains of AD patients leads to cognitive and behavioral changes, including memory loss, depression and anxiety [4]. AD affects multiple regions of the brain, including the hippocampus, subcortical nuclei, substantia nigra (SNR), ventral tegmental area (VTA) and the dorsal raphe [5]. The impact of AD in these areas may vary, including degeneration of afferent neurons, mitochondrial abnormalities and apoptosis of monoaminergic cells. The mechanism of the different effects is not currently well understood; however, it has been suggested that the lack of amyloid-beta clearance or its accumulation results in neurons becoming more susceptible to oxidative stress associated with aging, leading to impaired function [6].

Neurotransmission is the process by which a signal is conveyed between neurons via endogenous signaling molecules called neurotransmitters [7]. Neurotransmitters released from the axon terminal of one neuron cross the synaptic cleft and bind to receptors on the dendrites of another neuron, which are then converted into electrical signals. Synapse, the junction between neurons, has a tripartite structure that consists of presynaptic and postsynaptic nerve terminals along with the intimate association of glial cells [8]. Glial cells, in particular astrocytes, play an important role in regulating neurotransmission through feedback mechanism [8]. The binding of neurotransmitters onto specific postsynaptic receptors propagates signals from one neuron to another [7]. Postsynaptic receptors can be classified into metabotropic receptors that act through signal transduction and ionotropic receptors that act as ligand-gated channels [9]. Neurotransmitters can be classified into several types including monoaminergic and cholinergic neurotransmitters. In AD pathogenesis, important monoaminergic neurotransmitters include dopamine (DA), serotonin (5-HT) and histamine, whereas significant cholinergic neurotransmitters include gamma-aminobutyric acid (GABA) and acetylcholine (ACh) [5,10]. In AD, the therapeutic potential of neurotransmitter inhibitors has been well established, such as acetylcholinesterase inhibitors (AChEI) donepezil, rivastigmine, memantine and galantamine [9]. These drugs function by preventing the breakdown of ACh through inhibiting acetylcholinesterase (AChE) activity, which improves cholinergic transmission, inducing long-term potentiation in memory and learning. However, cholinesterase inhibitors only serve to alleviate the symptoms of AD and cannot halt the progression or reverse AD pathology. Interestingly, in recent years, melatonin has been proposed as an alternative neurotransmitter-based therapeutic drug.

Melatonin (N-acetyl-5-methoxytryptamine) is a monoaminergic tryptophan metabolite that is mainly synthesized in the pineal gland from its precursor serotonin. Melatonin is responsible for regulating the circadian rhythm, clearing free radicals, improving immunity and preventing oxidation of biomolecules. It has been reported that melatonin secretion gradually decreases with the progression of AD. Thus, it can be considered as a possible biomarker for AD [3]. Moreover, low levels of melatonin in AD patients have been observed to cause afternoon agitation, sleep disturbances and disrupted circadian rhythm, which further supports melatonin as a therapeutic target [11]. It has been shown that melatonin is able to inhibit the synthesis of Aβ plaques, prevent the formation of fibrils and protect neurons from the toxic effects of plaque accumulation [12]. In addition, melatonin was also found be effective in improving cholinergic and glutamatergic systems [13,14]. This is clinically significant as melatonin not only has low toxicity, but could also be an alternative monoaminergic drug in a saturated market of cholinesterase inhibitors [3]. However, only a small number of studies have investigated the effects of melatonin on various neurotransmitters or examined its effects compared to other cholinergic drugs, thus the efficacy of melatonin as a treatment for AD is currently not well established. However, the ability of melatonin to prevent the progression of AD pathology together with its neuroprotective traits suggest that melatonin could be a potential treatment for AD. In this review, we examine the potential of melatonin in AD treatment and prevention. We briefly describe the hallmarks of AD and summarize the protective effects of melatonin against these hallmarks, and then highlight the findings related to melatonin’s role in neurotransmission to understand the therapeutic benefits of melatonin in AD.

## 2. Major Pathologies Involved in AD

### 2.1. Role of Aβ Plaques and Neurofibrillary Tangles

Our current understanding of AD progression in terms of the development of abnormal structures such as Aβ plaques and NFT has been well documented. Amyloid-beta plaques consist of amyloid peptides of 36–43 amino acids with a molecular weight of about 4 kDa [1]. They are generated through the cleavage of amyloid precursor protein (APP), which is a type I membrane protein encoded by the *APP* gene on chromosome 21. Notably, a large proportion of familial Alzheimer’s patients have mutations in the *APP* gene. Moreover, those with Down syndrome are known to develop pathological and clinical indicators of early-onset AD, as an extra copy of chromosome 21 leads to the production of more APP. AD is a multifactorial disease as it can arise not only from mutations in the *APP* gene, but also from mutations in presenilin-1 (PSEN1) and presenilin-2 (PSEN2) genes. This is significant as presenilin is involved in the processing of APP, in which it acts as the catalytic subunit of gamma-secretase that cleaves the Aβ peptide from the carboxyl end of APP [15]. An alternative method of cleaving APP involves alpha-secretase, which generates soluble beta-APP (sAPP) that prevents accumulation of amyloidogenic fragments [1].

As our understanding of Aβ plaques has increased, the amyloid cascade hypothesis was put forward for the progression of AD. This hypothesis suggests that mutations in either *APP* or *PSEN* gene lead to the accumulation of amyloid peptides, which form into beta-pleated sheet structures, a process that occurs in familial AD. Alternatively, the hypothesis also proposes that sporadic AD may occur when the normal accumulation of Aβ peptides is sped up by external factors that reduce the clearance of these peptides. For instance, disrupted expression of aquaporin-4 (AQP4), which is expressed in the end-feet of astrocytes, impacts the glymphatic pathway, responsible for cleaning Aβ from the brain, hence facilitate the deposit of Aβ plaques [16,17,18,19]. Both pathways lead to the accumulation of Aβ peptides that oligomerize to form senile plaques, resulting in a series of downstream events including synapse degeneration, inflammation, plaque deposition, tau hyperphosphorylation and neuronal death [20].

### 2.2. Role of Aging

It is well established that the risk for AD increases with aging. The structural complexity of our brains gradually decreases during the normal aging process, including reduced complexity of neuronal dendrites, synapse density and astrocyte function [20]. It has been suggested that a series of events have occurred to trigger the development of AD. The first event involves an injury that initiates the pathogenic pathway, which could explain the tightly knit relationship between AD development and age. As our body’s ability to initiate a protective response in neurons begins to decline with age, any form of injury such as stroke or a traumatic brain injury may lead to the pathogenic development of AD [20]. The second event involves chronic inflammation that adds to the existing stress in the age-affected brain. The concept of chronic inflammation leading to AD has been well established. Studies have shown that prolonged use of non-steroidal anti-inflammatory drugs (NSAIDs) can reduce the risk of AD by 30–60% [21]. The third event involves the altered physiology of the neurons, which can occur during the prolonged inflammatory response. It was discovered that stressed neurons can re-enter the cell cycle, which halts after DNA replication and is unable to progress into late G2/M phase or revert the cell cycle progression. As a result, these neurons have duplicate copies of DNA content resulting in dramatic cellular changes [22], which ultimately lead to AD and dementia through disrupted synaptic dysfunction, neuronal loss and deficiencies in neurotransmission.

## 3. Dysfunction of Neurotransmission in AD

As mentioned above, it is well established that neurotransmission is affected in patients with AD, particularly the chemical neuroanatomy of monoaminergic systems including serotonergic and dopaminergic systems, cholinergic systems including acetylcholine and GABA, and glutamatergic systems [11]. Neurotransmission dysfunction mainly arises from the degeneration of neurons as a result of the toxic accumulation of Aβ plaques (see Table 1 and Figure 1).

### 3.1. Dysfunction of Cholinergic Neurotransmission

One of the most well-known neurotransmitters involved in AD is acetylcholine. Acetylcholine is a neurotransmitter that is widely distributed across the brain. The neurons responsible for ACh production are also widely distributed, innervating the majority of the brain’s neurons. Given the high distribution of ACh in the brain, it comes as no surprise that ACh is responsible for many physiological processes including stress response, memory and learning, which are all affected in AD [23]. The dysfunction in the neurotransmission of acetylcholine has been well documented in many studies. A study showed a negative correlation between choline acetyltransferase (ChAT), an enzyme in ACh synthesis and the degree of Aβ plaques in the brains of AD patients [24]. This finding was further supported by the discovery that the cerebral cortex had markedly reduced levels of presynaptic cholinergic markers, severe degeneration and the presence of NFT at the nucleus basalis of Meynert (NBM), which is a region responsible for ACh innervation [25]. Studies on TgAPP23 and 3xTg-AD mice showed a decrease ChAT level and its activity in the presence of Aβ plaques in the neocortex, basal forebrain and the hippocampus [26,27,28,29]. A study on Tg2576 mice showed an association between memory loss and decreased ACh release. They discovered that 9- to 11-months-old mice had lower levels of ACh released from the hippocampus accompanied by memory deficiencies [30]. A Similar result was also found in a study on 7 months old TgCRND8 mice exhibiting increased Aβ plaques, oxidative stress, inflammation and neurodegeneration [31]. In PDAPP mice, there were decreases of ChAT and AChE positive neurons [32]. In another study with the same model, reduced hippocampal ACh level found to be associated with hyper-locomotor function [33]. Interestingly, a study showed young APP/PS1 mice with cholinergic denervation had increased Aβ deposition accompanied by early memory impairments, suggesting a synergistic relationship between the progression of AD pathology and ACh denervation [34]. Overall, these findings indicate the decrease in ACh exacerbates or contributes to AD pathogenesis.

Another neurotransmitter involved in AD is GABA, which is the main inhibitory neurotransmitter in humans and acts to reduce the excitability of neurons throughout the nervous system. It is known to be responsible for cognition, memory, learning, motor function, circadian rhythm, neural development and adult neurogenesis [35]. Dysfunction in GABA is suspected to contribute to AD. However, the findings on the effects of GABA levels in AD vary from study to study. A study on both APP/PS1 and 5xFAD mice found that astrocytes could produce a large amount of GABA in the dentate gyrus of the hippocampus, whereas suppressing GABA production restored learning and memory [36]. In AβPPswe-PS1dE9 and TgCRND8 mice, accumulation of Aβ plaques and lower amounts of GABA were found in the striatum, hippocampus and cortex [37,38]. Nevertheless, an analysis of GABA levels in the human AD brain showed a general trend of either decreased or no change of GABA level in the brain regions such as hippocampus, subiculum, thalamus, subthalamic nucleus, cingulate cortex, amygdala and putamen [35]. A main concern of these studies is that they quantified GABA from post-mortem tissues. Post-mortem studies usually measure GABA by examining the activity of its synthesizing enzyme glutamic acid decarboxylase (GAD), but GAD is easily affected by pre-mortem environments, which can affect the measurements [39]. As such, more research is needed to understand the contribution of GABA levels and transmission in AD.

The glutamatergic system is tightly linked to GABA, given that glutamate is the precursor of GABA, which is converted into GABA by GAD [9]. However, in contrast with GABA, glutamate is an excitatory neurotransmitter. Glutamate is known to control cognition, memory and learning [40]. In AD, it has been shown there is a constant abnormal activation of a glutamatergic receptor, namely *N*-methyl-D-aspartate (NMDA) receptor [41,42]. NMDA receptors can be found within the extra-synaptic and synaptic regions. Overstimulation of the extra-synaptic NMDA receptor, for instance the GluN2B-composed NMDA receptors [43], leads to the loss of mitochondrial membrane potential, prolonged activation of extra-synaptic NMDA receptors, influx of Ca^2+^ ions, efflux of Mg^2+^ ions and excitotoxicity-induced neuronal loss [44]. This is in contrast to synaptic NMDA receptors, which are known to promote cell survival and synaptic activity [45]. In general, glutamate is released from the presynaptic terminals following an event of neuronal depolarization, and it activates different metabotropic and ionotropic receptors of postsynaptic, presynaptic neurons or even glial cells. Glutamate is then removed from the extracellular space by either excitatory amino acid transporters 1/2 (EAAT1/2) that located on the astrocytes or EAAT2/5 on the presynaptic terminal [46], which allows glutamate to be reused and secreted from the presynaptic neuron. EAATs are the primary mediators of excitatory transmission and glutamate transport in the CNS. Excitotoxicity occurs when glutamate release exceeds the reuptake of glutamate by EAATs, or when there is a decrease in EAAT expression, [47]. Studies have shown that in individuals with AD, Aβ increases glutamate availability by restricting the glutamate recycling mechanisms [48], thereby inducing glutamate release from astrocytes [49] and directly regulating the electrophysiological function of NMDA receptors [47]. It has been suggested that activation of the NMDA receptor increases Aβ production by inducing a shift from α-secretase to β-secretase, which alludes to overproduction of Aβ even in the condition of mild dysregulation of glutamatergic neurotransmission [47], and this leads to a vicious cycle of constant overstimulation of extra-synaptic NMDA receptors. This further triggers the signals for cell death and overwhelms the aforementioned synaptic NMDA receptor-mediated signals for cell survival [44]. Next, the balance is worsened by Aβ, in which damage of the glutamatergic neurotransmission, regulates the number of synaptic NMDA receptors through decreased expression and endocytosis of synaptic NMDA receptors, as well as limiting the synaptic plasticity [50]. Subsequently, the balance between synaptic and extra-synaptic NMDA receptors is disrupted, and this contributes to dysfunction of the glutamatergic system and its downstream signaling pathways that eventually leads to synaptic damage and neuronal death [44]. A study on AβPP/PS mice found significantly increased potassium-evoked glutamate release occurred in CA1 of the hippocampus prior to cognitive decline [51], suggesting that glutamate levels can act as an early biomarker for AD. However, this was contradicted in a study using AβPPswe-PS1dE9 mice, which found a significant decrease in cortical glutamate accompanied with a decrease in glucose, GABA and glutamate in the hippocampus and striatum when compared with wild-type mice [37]. Another study on PS2APP mice also found a significant reduction of glutamate in frontal cortex [52]. Abnormal function of glutamate receptor GluN2B and loss of metabotropic glutamate receptor 2 were also found in the same model [53,54]. Interestingly, an increase in the level of glutamate receptor GluN1 was observed in the neocortex, hippocampus and cerebellum, while increase of GluA2, was found only in the neocortex of TgCRND8 mice [55]. Importantly, mitochondrial damage-mediated excessive Ca*^2+^* influx in glutamatergic neurons also led to cholinergic system dysfunction by lowering choline acetyltransferase and ACh level. These results suggest that glutamatergic activity and its neurotransmitter cycle is impaired in AD. Given that the contrasting results could be due to variations in the study designs, more research is needed to improve our understanding of the glutamatergic system in AD.

### 3.2. Dysfunction of *Monoaminergic Neurotransmission*

The involvement of monoaminergic systems in AD has been well documented. Functional and structural changes of monoaminergic systems have been clearly shown to be involved in AD pathophysiology. In particular, the serotonergic system has been implicated in AD progression. Serotonin, also known as 5-hydroxytryptamine (5-HT), is a neurotransmitter produced from L-tryptophan by tryptophan hydroxylase. It regulates memory, learning, cognition, mood, sleep and other physiological processes [11]. Individuals with AD due to damaged serotonergic system have reduced 5-HT levels, resulting in altered mood, emotional expression and recognition; disrupted appetite and sleep-wake cycle; and confusion, agitation and depression [56]. A study on *APPswe/PS1*d*E9* mice with Aβ plaque accumulation showed they exhibited prominent monoaminergic neurodegeneration in forebrain 5-HT axons accompanied by anxiety-related behaviors at 18 months [57]. An animal study on Swiss mice also showed a disrupted serotonergic system, in which Aβ plaques were found to disturb the homeostatic regulation of 5-HT resulting in depressive-like behavior [58]. Another study on *APPswe/PS1*d*E9* mice showed a significant decrease in 5-HT2A receptor binding, which lowered serotonergic system activity [59]. In addition to reduced 5-HT secretions, it was shown that serotonergic denervation of the neocortex and hippocampus was accompanied by decreased production of 5-HIAA, a metabolite of 5-HT. Decreases in both 5-HT and 5-HIAA were found to be correlated with an increase in NFT, suggesting serotonergic systems were impaired in parallel with AD progression [60]. Furthermore, an interesting relationship between AD and the serotonergic system was found in the control of sleep-wake by the serotonergic DRN. In a study on *APPswe/PS1*d*E9* mice, plaque formation led to sleep-wake cycle deterioration [61]. However, it was also shown that lack of sleep caused poor clearance of Aβ plaques, resulting in more plaque formation that then exacerbates the sleep deprivation, forming a vicious cycle. Thus, the decrease in serotonergic activity in AD, in turn, leads to further disease progression, which illustrates the importance of the serotonergic system on AD development.

Another important monoaminergic system affected in AD is the dopaminergic system. Dopamine (DA) is responsible for the control of mood and multiple cognitive functions such as attention, working memory, thinking, social behaviors and learning [11]. It is synthesized from tyrosine by tyrosine hydroxylase [9] by dopaminergic neurons located in the mesencephalon and diencephalon, such as the substantia nigra pars compacta (SNc) and the VTA [62]. Dysfunction in the dopaminergic system in AD has been well documented in many studies. Reduction in DA, its metabolites and receptors have been observed in patients with AD [63]. In TgCRND8 AD mice, a decrease in the level of dopamine was found in the hippocampus, whereas, the increases of dopamine were observed in the frontal cortices and neostriata, suggesting dopaminergic dysfunction in AD [64]. A study on 5xFAD mice showed there was a significant decrease in both TH+ and TH- cells in DA-producing regions, and SNR-VTA networks were altered to enhance the synchronization of neuronal firing activity in DA-producing nuclei [65]. In addition, an animal study on a Tg2576 mouse model of AD found significant degeneration in VTA dopaminergic neurons compared to wild-type mice [66]. These results support the notion that neurological and cognitive deficits in AD are associated with disruptions in the dopaminergic system.

**Table 1 ijms-22-06841-t001:** In vivo studies related to neurotransmission in AD. Abbreviations*:* 5 HT, 5-hydroxy-tryptamine, Serotonin; ACh, Acetylcholine; AChE, Acetylcholinesterase; APPswe, Amyloid-beta precursor protein with Swedish mutation; Aβ, Amyloid-beta; AβO, Amyloid-beta oligomers; AβPP, Amyloid-beta precursor protein; ChAT, Choline acetyltransferase; DA, Dopamine; DAergic, Dopaminergic; GABA, Gamma-aminobutyric acid; MAergic, Monoaminergic; mGlu2, Metabotropic glutamate receptor 2; NA, Noradrenergic; NMDAR, N-methyl-D-aspartate receptor; PS, presenilin transgenic; SN, Substantia nigra; TH-, Tyrosine hydroxylase negative; TH+, Tyrosine hydroxylase positive; VTA, Ventral tegmental area.

Animal Model.	Gender	Age	Pathology Involved	Neurotransmission Dysfunction	Behavioral Effects	References
*APPswe/PS1*d*E9* mice	N/A	4–18 months old	Degeneration and loss of forebrain 5-HT and NA axons after Aβ deposits	Monoaminergic neurodegeneration	Anxiety-related behaviors in 18 months	[57]
Swiss mice treated with AβO	N/A	3 months old	Development of Aβ plaques	AβO disrupts 5-HT homeostasis	Depressive-like behavior	[58]
*APPswe/PS1*d*E9* mice	Male	4, 8, 11 months old	Progressive accumulation of Aβ protein.	Significant decrease in 5-HT2A receptor binding	Memory impairment	[59]
5xFAD mice	Male	6 months old	Significant decrease of both TH+ and TH- cells in DA-producing areas	SN-VTA networks are enhanced to the synchronization of neuronal firing activity in DA-producing nuclei	Cognitive malfunction Synaptic malfunction	[65]
Tg2576 mice	Male	2 and 6 months old	Degeneration of VTA DAergic neurons	Reduced noradrenergic transmission in dorsal subiculum	Age-related impairment of memory and non-cognitive functions	[66]
Tg2576 mice	N/A	4–6 and 9–11 months old	Aβ were prominent in 20-month-old mice	Reduced ACh release from hippocampus in 9- to 11-month-old mice	Memory impairment present in 9- to 11-month-old mice	[30]
APP/PS1 mice	N/A	3 and 7 months old	Aβ plaques deposition after cholinergic degeneration	Dramatically reduced cholinergic neurons Neuronal loss in nucleus basalis	Early memory impairment Progressive impairment	[34]
APP/PS1 and 5xFAD mice	N/A	8 and 13 months old	Aβ plaques deposition and reactive astrocytes	Aberrant increase in GABA release from reactive astrocytes	Impaired learning and memory	[36]
AβPP/PS mice	Male	2–4 months old	Abnormal glutamate release precedes cognitive decline	Significantly increased potassium-evoked glutamate release in CA1	Cognitive decline	[51]
*A*βPPswe-PS1dE9 mice	N/A	6 months old	Deposition of Aβ plaques	Significant decrease in cortical glutamate and GABA Glucose, GABA and glutamate reduced in hippocampus and striatum	Impairment of cognitive function and memory	[37]
TgAPP23 mice	Male and female	24 months old	Deposition of Aβ plaques and cholinergic degeneration	Decreased ChAT-positive boutons in neocortex Significant reduction of ChAT-positive neurons volume in basal forebrain	N/A	[26]
PS2APP mice	Female	20 or 24 months old	Deposition of Aβ plaques	Significant reduction of glutamate level in frontal cortex	N/A	[52]
TgAPP23 mice	N/A	7–8 months old	Dysfunction of cholinergic and monoaminergic systems	Decreased AChE and ChAT activity in basal forebrain nuclei Increased 5-HT levels in parietal cortex and occipital cortex	N/A	[27]
PDAPP mice	Male and female	4–6 months old	Deposition of Aβ plaques	Reduced basal and evoked ACh release from hippocampus	Hyper-locomotor function	[33]
3xTg-AD mice	Male and female	2–4, 13–15 and 18–20 months old	Aβ plaques deposition with cholinergic degeneration and alteration of neurotrophic factors	Reduced ChAT in medial septum/vertical limb of the diagonal band of Broca in 18- to 20-month-old mice Decreased hippocampal ChAT activity in 13- to 15-month-old mice	N/A	[28]
hAPP-J20 mice	N/A	6 months old	Altered synaptic plasticity and cognitive function	Significantly decreased phospho GluN2B levels and hippocampal LTP	Impaired learning and memory	[42]
TgCRND8 mice	N/A	2 and 7 months old	Aβ plaques deposition, oxidative stress, reactive glial cells and neurodegeneration	Reduced ChAT-positive neurons and ACh levels.	Cognitive impairment	[31]
PS2APP mice	Male	5, 9, 13 and 17 months old	Deposition of Aβ plaques	Significant loss of mGlu2 receptors in entorhinal cortex and lacunosum moleculare regions	N/A	[53]
PS2APP mice	Male	3–4 months old	Altered synaptic plasticity	Aberrant GluN2B-NMDAR function	N/A	[54]
PDAPP mice	Male	2, 4, 12 and 24 months old	Aβ plaques deposition with cholinergic degeneration	Reduced Cholinergic nerve terminals density Significantly decreased ChAT activity	N/A	[32]
3xTg-AD mice	N/A	9–23 months old	Deposition of Aβ plaques	Reduced ChAT and AChE-positive neurons	N/A	[29]
TgCRND8 mice	Male	3 months old	Deposition of Aβ plaques and neuronal degeneration	Significantly increased GluN1 in neocortex, hippocampus and cerebellum. Significantly increased GluA2 in neocortex but decreased in hippocampus	Cognitive impairment	[55]
TgCRND8 mice	Male and female	2–3 and 12–13 months old	Deposition of Aβ plaques	Decreased glutamate in hippocampus, cortex, frontal cortex and midbrain Decreased GABA in hippocampus, cortex and midbrain	N/A	[38]
TgCRND8 mice	Male	3 months old	Dysfunction of dopaminergic system	Increased dopamine level in the neostriata and frontal cortices Decreased dopamine level in the hippocampus	Cognitive impairment	[64]

## 4. Role of Melatonin against AD Hallmarks

As serotonin is the precursor of melatonin, it stands to reason that melatonin would be closely associated to the pathogenesis of AD. Evidence from various preclinical studies support that melatonin has therapeutic effects against AD (see Table 2 and Table 3). Melatonin has been shown to ameliorate the effects of AD in multiple ways, primarily in the processing of Aβ plaques and in reducing tau hyperphosphorylation [3,67].

### 4.1. Anti-Amyloidogenic Effects

Melatonin was found to be able to disrupt the histamine-aspartate salt bridges in Aβ peptides, resulting in destabilization of the beta-sheet structures and leading to a large difference in beta-sheet content between Aβ plaques incubated with and without melatonin [68]. Similar to other anti-amyloidogenic substances, melatonin is able to inhibit aggregation of Aβ plaques to a certain degree, as revealed through electrospray ionization mass spectrometry (ESI-MS) [69]. Many studies support melatonin’s ability to reduce Aβ plaque levels in the brain. For instance, a study on AβPP/PS mice found that melatonin was able to reduce the amount of amyloid plaques in the hippocampus and frontal cortex, which was accompanied by improved spatial learning and memory [70,71]. Additionally, it was reported that melatonin was able to inhibit the expected time-dependent development of Aβ plaques in Tg2576 mice [72]. Another study showed that melatonin was unable to clear plaques or prevent further Aβ plaque generation in older Tg2576 mice [73]. Interestingly, melatonin was also found the ability to increase the removal of Aβ by facilitating the glymphatic system [74,75]. A clinical study has shown that the elimination of Aβ in sleeping brain is drastically increased compared to waking brain [74]. It was further proved in a study on Tg2576 mouse model, where melatonin treatment was able to augment the clearance of Aβ [75]. These findings suggest that melatonin can successfully prevent Aβ pathology and modulate APP metabolism, but is unable to exert anti-amyloidogenic effects once there is severe Aβ plaque deposition. This supports that melatonin may be more effective as a preventive drug rather than for treating the effects of AD. However, the mechanisms underlying the ability of melatonin to reduce Aβ plaque levels are not clear.

### 4.2. Inhibition of Tau Hyperphosphorylation

The ability of melatonin to inhibit tau hyperphosphorylation has been well documented in multiple in vitro and in vivo studies. For example, studies on N2a neuroblastoma cells with hyperphosphorylated tau induced by either calyculin-A [76] or wortmannin [77] showed that melatonin could maintain cell viability by inhibiting tau hyperphosphorylation. In a study on 3xTg-AD mice, melatonin was shown to decrease the level of hyperphosphorylated tau, and together with exercise was also able to decrease the level of Aβ oligomers [78]. In the same study, melatonin was able to protect against cognitive impairments, brain oxidative stress and decreased mitochondrial DNA. In addition, a study on ICR mice injected with Aβ plaques to induce AD showed they had decreased expression of hyperphosphorylated tau that resulted in improved neuron viability [79]. Furthermore, the inhibition of the melatonin synthesizing enzyme 5-hydroxyindole-O-methyltransferase was found to cause tau phosphorylation and spatial memory impairments, which were reversed by injection of melatonin supplementation for 1 week [80]. These findings indicate that melatonin can ameliorate the symptoms of AD through arresting tau hyperphosphorylation.

**Table 2 ijms-22-06841-t002:** In vitro studies related to the neuroprotective effects of melatonin in AD. Abbreviations: C6, Rat glial cell; HeLa, Immortal human cell line; N1E-115, Mouse neuroblastoma cell line; PC12, Rat pheochromocytoma cell line; sAPP, Soluble derivates of Amyloid-beta precursor protein; SHSY5Y, Human neuroblastoma cell line; SK-N-SH, Human neuroblastoma cell line; SV770, Monkey kidney cell line; U-138, Human astrocytic cell line.

Study Model	Effects on AD Pathology	References
Multiple cell types (SK-N-SH, SHSY5Y, U-138, SV770, C6, PC12, N1E-115)	Decrease in soluble APP secretion in PC12, SV770, U-138, HeLa, N1E-115	[1]
N2a neuroblastoma cell	Protective effects against tau hyperphosphorylation induced by wortmannin	[77]
N2a neuroblastoma cell	Protective effects against tau hyperphosphorylation induced by calyculin-A	[76]

**Table 3 ijms-22-06841-t003:** In vivo studies related to the neuroprotective effects of melatonin in AD. *Abbreviations:* AChE, Acetylcholinesterase; APP, Amyloid-beta precursor protein; Aβ, Amyloid-beta, AβO, Amyloid-beta oligomers; BDNF, Brain-derived neurotrophic factor; ChAT, Choline acetyltransferase; CREB, cAMP Response Element-Binding Protein; LPS, Lipopolysaccharides; SD, Sprague Dawley.

Animal Model	Gender	Age	Treatment Dosage and Duration	AD Pathology Involved	Effects on AD Pathology	Effects on Neurotransmission	References
Tg2576 mice	N/A	8, 9.5, 11, 12.5 months old	0.5 mg/mL, 4, 5.5, 7, 8.5 months	Plaque-like deposits of amyloid-beta	Reduced time-dependent Aβ levels, abnormal nitration on proteins Increased mice survival	N/A	[72]
ICR mice treated with Aβ_1-42_	Male	N/A	10 mg/kg, 5 mg/kg, 2.5 mg/kg, 14 days	Affected cognitive functions	Improved cognitive deficits and spontaneous activity of mice Reduced hyperphosphorylated tau expression	Improved neuron viability	[79]
AβPP/PS mice	N/A	4 months old	100 µg/mL,0.5 mg/day	Amyloid plaques, behavioral deficits	Melatonin improved spatial learning and memory Aβ load in hippocampus and frontal cortex were reduced	N/A	[44]
APP/PS1 mice	N/A	2–2.5 months old	100 mg/L	Aβ plaques	Reduced Aβ plaques deposition in hippocampus and entorhinal cortex Decreased inflammatory cytokines in hippocampus	N/A	[71]
3xTg-AD mice	Male	6 months old	10 mg/kg body weight	AβO, hyper-phosphorylated tau	Decreased number of Aβ oligomers and hyperphosphorylated tau Protection from cognitive impairment, brain oxidative stress, decrease in mitochondrial DNA	N/A	[78]
SD rats treated with LPS	N/A	N/A	5 and 10 mg/kg	Inflammation, oxidation, increased AChE activity	Lowered levels of induced inflammation and oxidation	Inhibited increase in AchE activity	[13]
APP695 mice	N/A	4 months old	10 mg/kg/day	Aβ plaques, decreased ChAT levels	Long-term treatment significantly reduced Aβ plaque levels	Increased ChAT activity in frontal cortex and hippocampus	[81]
Swiss mice treated with AlCl_3_ and d-galactose	Male	N/A	80 mg/kg/day	Affected cognitive functions, decreased BDNF, CREB and AChE levels	Increased BDNF and CREB levels Improved memory deficits	Increased AChE level	[82]

## 5. Role of Melatonin on Neurotransmission

The effects of melatonin on neurotransmission primarily involve improvements in cholinergic and glutamatergic systems (see Figure 2). As mentioned previously, Aβ plaques can impair the function of glutamatergic neurons and cause excessive influx of calcium, which lead to overstimulation and unnecessary release of AChE, resulting in reduced choline acetyltransferase and ACh levels.

Melatonin has been hypothesized to alleviate the disruption of the cholinergic system in AD through inhibiting the calcium-induced release of AChE, thus effectively acting as an acetylcholine enhancer [3]. Supporting this hypothesis, a study on a sporadic AD rat model showed that melatonin treatment could significantly decrease the level of inflammation and oxidation, as well as inhibit AChE activity [13]. In addition, it has been shown that as AD progresses, choline acetyltransferase [25] and AChE synthesis begins to decrease, which correlates positively with dementia severity in AD patients. Melatonin has also been shown to promote choline transport, which improved ACh synthesis [69]. In APP695 mice, melatonin treatment significantly decreased ChAT activity in the frontal cortex and hippocampus [81]. In a recent study on sporadic AD mice, melatonin rescued the AChE level and promoted neuroprotection [82].

Melatonin has also been suggested to alleviate the altered glutamatergic system in AD by inhibiting the activity of NMDA receptors. Melatonin was able to reduce excessive Ca^2+^ influx by altering the activity of voltage-gated Ca^2+^ channels, thereby inhibiting the effects of NMDA receptors [83]. This was supported by a study on adult male Wistar rats, which found that melatonin treatment attenuated the glutamatergic-dependent excitatory response in striatal neurons by reducing Ca^2+^ influx in voltage-gated Ca^2+^ channels and NMDA-gated Ca^2+^ channels, resulting in an anti-excitotoxic effect [14]. Even though these findings show melatonin has beneficial effects on cholinergic and glutamatergic systems, the impact of melatonin on the neurotransmission of monoaminergic systems has yet to be demonstrated, which will require more research.

## 6. Conclusions and Future Perspective

The effects of melatonin on neurotransmission and AD pathologies have been separately investigated in several respective studies. Melatonin is well established as a therapeutic for sleep disorders and jet lag, and has been investigated as an adjunct medication for cancer patients and as a medication for free-radical diseases. In this review, we have explored how melatonin can serve as a therapeutic for AD by inhibiting the pathological progression and restoring cholinergic and glutamatergic neurotransmission. Nevertheless, more research is needed to reveal its effects on other neurotransmitters such as GABA, serotonin, dopamine and histamine. Apart from the study on the direct interaction of melatonin with neurotransmission, the pathway through which melatonin can indirectly clear Aβ plaques is also worth studying since Aβ plaques are the fundamental source to cause neurotransmission dysfunction, and melatonin can reinforce the clearing effect of the glymphatic pathway by utilizing melatonin-AQP4 interaction. In addition, recent preclinical studies have indicated that melatonin metabolite N(1)-Acetyl-N(1)-formyl-5-methoxykynuramine, melatonin-derived benzylpyridinium bromides, melatonylvalpromide and melatonin-N,N-Dibenzyl(N-methyl) amine hybrids have neuroprotective effects against AD pathologies [84,85,86,87]. Moreover, melatonin receptor agonists piromelatine and agomelatine were found be effective against AD [88,89], but their neuroprotective effects against neurotransmission dysfunction in AD are still unknown. Further preclinical research is needed to investigate the detailed role of these compounds on different neurotransmission systems in AD, and further clinical studies on melatonin and its compounds are needed to validate their efficacy in the different stages of AD. To facilitate future studies, the recent advancement of neurotransmitter imaging techniques including positron emission tomography (PET) and Single-Photon Emission Computed Tomography (SPECT) can be taken into consideration. These are useful for accurate real-time neurotransmission detection. Furthermore, the improvement of the melatonin delivery system and the genetic variation of melatonin response can be evaluated by high-throughput screening and computer-aided drug design [90,91]. Finally, by understanding the mechanisms of how melatonin ameliorates AD pathogenesis, we can further ascertain its therapeutic value. Without a doubt, the future of melatonin as a potential treatment for AD is bright.

## Figures and Tables

**Figure 1 ijms-22-06841-f001:**
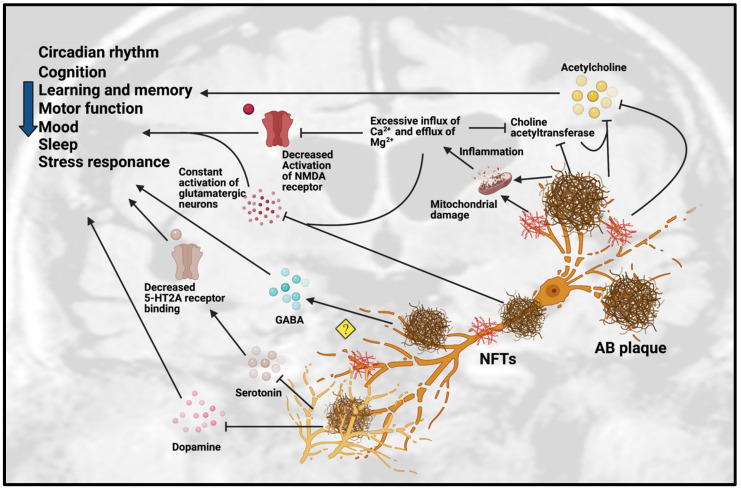
Dysfunction of neurotransmission in AD. Accumulation of Aβ plaques and NFT in AD cause impairment of the circadian rhythm, cognition, learning, memory, motor function, mood, sleep and stress response. These pathologies are toxic to neurotransmission systems, affecting cholinergic, glutamatergic, serotonergic and dopaminergic systems. Amyloid-beta plaques and NFT can inhibit the release of ACh and choline acetyltransferase, an enzyme that regulates ACh synthesis, which reinforces the inhibition effect of ACh. Amyloid-beta plaques and NFT can cause mitochondrial damage in glutamatergic neurons. The mitochondrial damage leads to inflammation due to excessive influx of Ca^2+^ and excessive efflux of Mg^2+^ that affect the activation of glutamatergic neurons and decreases the activation of NMDA receptor. The excessive influx of Ca*^2+^* in glutamatergic neurons leads to inhibition of choline acetyltransferase and further inhibits the synthesis of ACh. However, the detailed mechanisms are not yet understood, as some studies showed the upregulation of GABA in certain regions but downregulation of GABA in other regions. Amyloid-beta plaques also disrupt the homeostatsis of serotonin (5-HT) by inhibiting the binding of serotonin receptor (5-HT2A) and disrupting the dopaminergic system. Abbreviations: Aβ, Amyloid-beta; NFT, neurofibrillary tangle.

**Figure 2 ijms-22-06841-f002:**
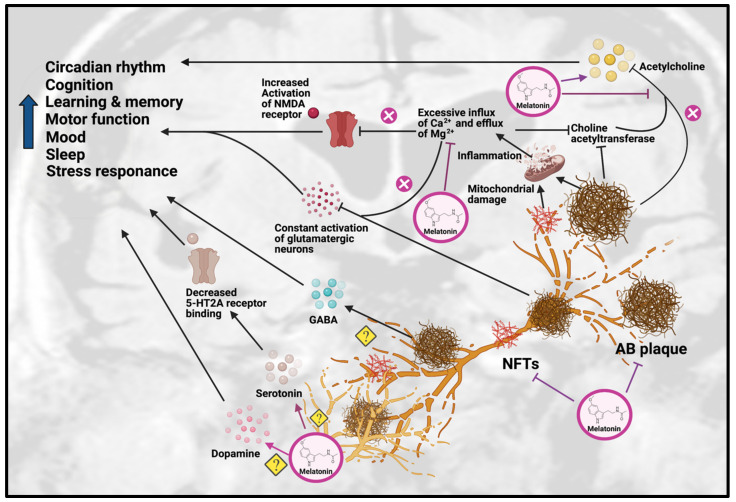
Effects of Melatonin treatment on dysfunction of neurotransmission in AD. Melatonin can ameliorate the formation of Aβ plaques and NFT, as well as improve the impairments due to these AD hallmarks, including disrupted circadian rhythm, cognition, learning, memory, motor function, mood, sleep and stress response. Melatonin treatment can have beneficial effects on serotonergic and dopaminergic systems, but the exact mechanisms have yet to be determined. Melatonin can also have beneficial effects on the cholinergic system by increasing acetylcholine release and reducing inflammation caused by excessive influx of Ca^2+^ and excessive efflux of Mg^2+^, thereby inhibiting choline acetyltransferase. Abbreviations: Aβ, Amyloid-beta; NFT, neurofibrillary tangle.

## Data Availability

Not applicable.

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
