# Peer review of "Regulation of Melatonin and Neurotransmission in Alzheimer’s Disease"

_ijms, 2021, doi:10.3390/ijms22136841_

Round 1

Reviewer 1 Report

Ng et al have summarised pathologies and dysfunctions in AD to then focus onto the possibile role of melatonin in these mechanisms and neurotrasmission exploring its potential as neurotherapeutic.

The review is well written and discussed; however, i believe it needs further work.

  1. In the paragraph, 3.1 (lines 155-160), please be more specific when talking about GABA levels and th differences between brain regions.
  2. in the analysis of the AD animal models, not all of them are mentioned, i.e no examples of PS2/APP mouse model. Please review again all the AD models in literature.
  3.  I would also add a paragraph, going more in depth how overactivation of the glutamate receptors (mentioned in 3.1 line 170) has been shown to be important in activating a vicious cycle with calcium overload and mitochondrial dysfunctions in AD.

Reviewer 2 Report

Dear Editor,

The manuscript by Ng and Roy et al discussed the effect of Alzheimer’s on neurotransmission the potential roles of melatonin in enhancing neurotransmission and its impact on the pathology of Alzheimer’s disease.

The review is comprehensive, informative and up-to-date (in most parts). Authors were successful in providing some well compiled opinions and summaries. The mechanistic figures and tables can be a good starting point for future studies and will be of interest for IJMS readers and beyond.

However, there is a number of major and minor points that would need to be addressed in order to improve the quality of this paper before it can be accepted for publication.

Major:

- This review overlooked some essential and up-to-date work regarding the pathophysiology of AD and neurodegenerative diseases (NDs), especially the role of glial cells and recent advances in their target validation and future therapies for NDs.

Authors should start by mentioning the role of glial cells (particularly astrocytes) in tripartite synapse in neurotransmission in lines 37-38 “Neurotransmission is the process by which a signal is conveyed between neurons via endogenous signaling molecules called neurotransmitters”. References to be included:

https://pubmed.ncbi.nlm.nih.gov/10322493/

https://pubmed.ncbi.nlm.nih.gov/31318452/

Neurodegenerative diseases are no longer neuron only disorders since the role of glial cells have been extensively validated. For example, the water channel aquaporin 4 (AQP4), which is highly expressed on astrocyte endfeet, critically regulates water flux between blood and brain. AQP4 plays an essential role in regulating the recently discovered glymphatic system which is a waste clearance system that utilizes a unique system of perivascular channels, formed by astroglial cells, to promote efficient elimination of soluble proteins and metabolites from the central nervous system. References:

https://www.ncbi.nlm.nih.gov/pmc/articles/PMC4252540/

https://pubmed.ncbi.nlm.nih.gov/30561329/

https://pubmed.ncbi.nlm.nih.gov/33004510/

Lines of evidence showed that CNS disorders NDs caused by astrocytes through AQP4 disorganization, leading to brain edema and exacerbate neurodegenerative diseases. This has been recently been demonstrated by the breakthrough study by Kitchen et al Cell 2020 where they showed that targeting AQP4 following ischemia and hypoxia not only reduces edema but also stabilises the BBB/BSCB barriers and led to accelerated functional recovery compared with untreated animals. This role has been recently been confirmed by the work of Sylvain et al BBA 2021 which has demonstrated that targeting AQP4 is a viable therapeutic target using a photothrombotic stroke model. They have also shown a link to brain energy metabolism as indicated by the increase of glycogen levels. References to be included:

https://pubmed.ncbi.nlm.nih.gov/32413299/

https://pubmed.ncbi.nlm.nih.gov/33561476/

Moreover, authors need to mention the work by Qian et al. Nature 2020 where they have beautifully shown that the conversion of midbrain astrocytes to dopaminergic neurons, which provide axons to reconstruct the nigrostriatal circuit. References:

https://pubmed.ncbi.nlm.nih.gov/32581380/

- Author needs to briefly discuss future directions following towards the end of their discussion and conclusion. This could include, but not limit to, the use of humanized self-organized models, organoids, 3D cultures and human microvessel-on-a-chip platforms especially those which are amenable for advanced imaging such as TEM and expansion microscopy since they enable real-time monitoring of neurotransmission. References to be included:

https://pubmed.ncbi.nlm.nih.gov/30165870/

https://pubmed.ncbi.nlm.nih.gov/33117784/

https://pubmed.ncbi.nlm.nih.gov/33180261/

-End of discussion and towards the conclusion: NDs are yet incurable diseases. Author needs to point out to the recent advances in applying the use of high-throughput screening and computer-aided drug design as have been nicely reviewed by Aldewachi et al 2021 and Salman et al 2021 as they can provide a novel insight that can support target validation in future studies. References to be included:

https://pubmed.ncbi.nlm.nih.gov/33672148/

https://pubmed.ncbi.nlm.nih.gov/33925236/

Minor:

- Define abbreviations whenever they appear first in the manuscript and use them throughout. For example, AD in line 11 should be introduced in line 9.

Best.

Round 2

Reviewer 1 Report

The authors have only partially addressed my concerns.

Only one example of paper for PS2APP mouse model and a very brief description of NMDA excitotoxicity in AD has been mentioned please review the literature more in detail.

Author Response

Please, see attached file.

Reviewer 2 Report

Dear Editor,

The authors have successfully addressed the majority of my comments and concerns in order to improve the quality of the manuscript.

I believe that the corrections, additional sections and updated references, have contributed to enhancing the clarity of the manuscript. I can endorse this manuscript for publication following the successful addressing of the minor edits below.

All the best!

Minor:

-Section 2.1: the references regarding the differential expression of AQP4 at the endfeet of astrocytes and their role in CNS (patho-)physiology are missing. References:

https://pubmed.ncbi.nlm.nih.gov/32413299/ 

https://pubmed.ncbi.nlm.nih.gov/33561476/

-Section 6, lines 385-386: the suggested references for HTS and CADD are missing. References:

https://pubmed.ncbi.nlm.nih.gov/33672148/ 

https://pubmed.ncbi.nlm.nih.gov/33925236/
